# A Strategical Improvement in the Performance of CO_2_/N_2_ Gas Permeation via Conjugation of L-Tyrosine onto Chitosan Membrane

**DOI:** 10.3390/membranes13050487

**Published:** 2023-04-29

**Authors:** Aviti Katare, Rajashree Borgohain, Babul Prasad, Bishnupada Mandal

**Affiliations:** 1Department of Chemical Engineering, Indian Institute of Technology Guwahati, Guwahati 781039, Assam, India; 2William G. Lowrie Department of Chemical and Biomolecular Engineering, The Ohio State University, Columbus, OH 43210-1350, USA

**Keywords:** CO_2_/N_2_ separation, chitosan, L-tyrosine, grafting, biopolymer, amino acid

## Abstract

Rubbery polymeric membranes, containing amine carriers, have received much attention in CO_2_ separation because of their easy fabrication, low cost, and excellent separation performance. The present study focuses on the versatile aspects of covalent conjugation of L-tyrosine (Tyr) onto the high molecular weight chitosan (CS) accomplished by using carbodiimide as a coupling agent for CO_2_/N_2_ separation. The fabricated membrane was subjected to FTIR, XRD, TGA, AFM, FESEM, and moisture retention tests to examine the thermal and physicochemical properties. The defect-free dense layer of tyrosine-conjugated-chitosan, with active layer thickness within the range of ~600 nm, was cast and employed for mixed gas (CO_2_/N_2_) separation study in the temperature range of 25−115 °C in both dry and swollen conditions and compared to that of a neat CS membrane. An enhancement in the thermal stability and amorphousness was displayed by TGA and XRD spectra, respectively, for the prepared membranes. The fabricated membrane showed reasonably good CO_2_ permeance of around 103 GPU and CO_2_/N_2_ selectivity of 32 by maintaining a sweep/feed moisture flow rate of 0.05/0.03 mL/min, respectively, an operating temperature of 85 °C, and a feed pressure of 32 psi. The composite membrane demonstrated high permeance because of the chemical grafting compared to the bare chitosan. Additionally, the excellent moisture retention capacity of the fabricated membrane accelerates high CO_2_ uptake by amine carriers, owing to the reversible zwitterion reaction. All the features make this membrane a potential membrane material for CO_2_ capture.

## 1. Introduction

The fossil fuel combustion method remains the biggest energy source for the planet’s needs [1]. The sharp increase in energy demand in the previous years is responsible for almost 33 Gt CO_2_ emissions in 2019, of which almost 30% comes from coal-fired energy generation plants [2]. In the pre-industrialized era, plants were the natural reservoir of carbon. They utilized CO_2_ via a process called photosynthesis. However, the transition of society from the preindustrial to the industrial stage, as well as deforestation and a rapid rise in energy demand due to population growth, have greatly increased the CO_2_ concentration levels [3,4]. Therefore, various technologies have been developed, investigated, and patented over the past few decades, comprising adsorption, absorption, cryogenic fractionation, and membrane technology for CO_2_ capture [5,6]. Membrane technology has gained interest as the preferred alternative to CO_2_ capture over more sophisticated absorption technology due to its operational simplicity, product consistency with high selectivity and CO_2_ loading, energy efficiency, its economical properties, and easy handling [7].

Although many materials have been used to prepare gas separation membranes, polymers have emerged as the primary market due to their simplicity of fabrication, low manufacturing costs, and fascinating mechanical properties [8,9]. So far, various polymeric materials have been successfully utilized, such as cellulose acetate [10,11,12], polyvinyl alcohol (PVA) [13,14,15], chitosan [16,17,18], polysulfones (PSf) [19], polyether-block-amide (PEBAX) [20,21,22,23], polyimides [24], polyvinyl amine [12,25,26,27,28], melamine [29], etc. to separate different binary gaseous mixtures, such as flue gas (CO_2_/N_2_), natural gas (CO_2_/CH_4_), oxyhydrogen gas (H_2_/O_2_), etc., as well as tertiary gaseous mixtures, such as CO_2_/N_2_/H_2,_ etc. [10,30,31].

Using aqueous solutions of monoethanolamine (MEA) and 2-amino-2-methyl-1-propanol (AMP), deMontigny et al. evaluated the performance of microporous polypropylene (PP) and polytetrafluoroethylene (PTFE) hollow fiber membranes in a system and discovered a CO_2_ flux of 0.1 × 10^−4^ mol/(m^2^ s). The groups found that, despite having a significantly higher wall thickness and a lower module specific area than the PP membranes, the 2-mm PTFE membranes performed better than the PP membranes. While the PTFE membranes maintained their level of performance throughout the testing period, the PP membranes demonstrated a decline in performance over time [32]. Hwang et al. examined the CO_2_/N_2_ separation capabilities of a commercialized polysulfone (PSf) hollow-fiber membrane module to be utilized for post-combustion CO_2_ capture. The CO_2_ permeance and CO_2_/N_2_ selectivity for the pure gas test were obtained 120 GPU and 26.4, respectively, and it was ascribed that the membrane separation performance was closely correlated with the instinct selectivity of PSf, suggesting that separation may take place at the thin skin layer of PSf [33]. Gomez-Coma et al. utilized [emim][Ac] ionic liquid and a commercial polysulfone hollow fiber membrane contractor for CO_2_ capture application and obtained a CO_2_ capture efficiency of 45% at 75 °C [34].

Among biopolymers, chitosan emerged as a highly interesting membrane material for CO_2_ separation [35]. The transport of carbonic acid gas and class of hydroxyl radicals across the membrane is obtained by high primary amine-containing chitosan. Membranes, in general, depend on the solution–diffusion mechanism and act as a molecular sieve based on the kinetic diameter and condensing ability of the various molecules [36]. The membrane performance efficacy is obtained by optimizing and tuning between the CO_2_ permeability and CO_2_/N_2_ selectivity, as discussed by Robeson’s “upper-bound” [37]. Researchers have utilized several methods to minimize this “trade-off” by modifying the existing polymer, such as blending [38], crosslinking [14], filler addition [39,40], thermal/chemical grafting [41], etc. The main emphasis of utilizing these strategies is to develop a facilitated transport mechanism (FTM). The diffusion of the target gas molecules, such as CO_2_, comprises the solution diffusion mechanism along with a reversible reaction with a functional group (called carriers) in the membrane structure.

The Zwitterion mechanism explains the reaction between CO_2_ and undisturbed carrier amines, described by Caplow as [42]
(1)$${CO}_{2}+2{RNH}_{2}⇌{RHNCOO}^{-}+{RNH}_{3}^{+}$$
(2)$${CO}_{2}+2RR^{\prime }NH⇌2{RR^{\prime }NCOO}^{-}+RR^{\prime }{NH}_{2}^{+}$$
(3)$$2{CO}_{2}+2{RNH}_{2}+H_{2}O⇌RHNCOOH+{RNH}_{3}^{+}+{HCO}_{3}^{-}$$
(4)$$2{CO}_{2}+2RR^{\prime }NH+H_{2}O⇌RR^{\prime }NCOOH+RR^{\prime }{NH}_{2}^{+}+{HCO}_{3}^{-}$$

Reactions (1) and (2) suggest that the CO_2_ intake is 0.5 mol of CO_2_/mol of amine. The presence of H_2_O molecules also plays a vital role in CO_2_ loading (Reactions (3) and (4)). Hence, the theoretical maximum CO_2_ intake is achieved in the presence of moisture.

Therefore, to improve the CO_2_ separation efficiency and overcome the Robeson “upper bound” trade-off, CS can be covalently conjugated or blended with various amines/polymers, such as poly(allylamine) (PAA), tetraethylenepentamine (TEPA), polyethyleneimine (PEI), polyvinyl amine (PVAm), etc. [16,43,44]. Solid-state FTM is developed by blending carboxymethyl chitosan (CMC) with carrier poly(amidoamine), resulting in 100 GPU CO_2_ permeance and 149 CO_2_/N_2_ selectivity. This strategy gave an advantage of ~89% improvement in CO_2_ permeance and ~64% in CO_2_/N_2_ selectivity compared to pristine CMC for binary gas (20% CO_2_ + 80% N_2_), respectively, at 90 °C [45]. Another class of membrane was fabricated by grafting PEG onto CNT and mixed with PEBAX 1657. The hybrid membrane showed CO_2_ permeability and CO_2_/N_2_ selectivity as a 369.1 barrier and 110 barrier, respectively, which overcomes Robeson’s upper bound limitation [46]. It is believed that the induced amorphous nature and functionalization through PEG grafting on CNT improved the available fractional free volume and tortuosity, which enhanced the reactivity/affinity towards CO_2_. Among many approaches, chemical grafting has been the most promising approach to enhance the CO_2_ separation performances of polymers [41,47].

An amino acid, L-tyrosine (Tyr), associated with phenolic hydroxyl groups, has stood out as an important biomaterial due to its peculiar biocompatible and mechanical properties. It is used as a precursor for food compounds, pharma, chemical, and cosmetic applications [48,49,50]. So far, the collective impact of the chitosan–tyrosine conjugate on the CO_2_ separation performance has remained unexplored. The key novelty of this research study is the successful conjugation of L-tyrosine onto chitosan, which has greatly increased the amorphousness of the membrane and proven a good membrane material for the effective separation of CO_2_ from flue gas. Apart from this, the influence of important operating variables, such as temperature, feed side moisture, sweep moisture, and transmembrane pressure, is precisely investigated as another novelty of this research paper.

## 2. Experimental Section

### 2.1. Requisite Materials

L-3-[4-hydroxyphenyl] alanine (L-tyrosine) was acquired from HiMedia Laboratories Pvt. Ltd. (Mumbai, India). Chitosan flakes (CS) with 310−375 kDa molecular weight and 1-hydroxy benzotriazole hydrate (HOBt) were provided by Sigma-Aldrich (Pompano Beach, FL, USA). 1-(3-dimethyl aminopropyl)-3-ethyl carbodiimide hydrochloride (EDC-HCl, 99%) was acquired from Spectrochemical Pvt. Ltd. (Mumbai, India). Acetic acid (CH_3_COOH, ≥99%) and hydrochloric acid (HCl, ~37% concentrated) were purchased from Merck Pvt. Ltd. (Mumbai, India). The feed gas mixtures containing 20 vol% CO_2_ + rest N_2_, helium gas (99.999 vol% purity), air-gas, and argon gas were kindly supplied by Vadilal Chemicals Ltd. (Vadodara, India). Poly (ether sulfone) (PES) with a thickness and mean pore size of 150 µm and 0.03 µm, respectively, were provided by Sterlitech, (Auburn, WA, USA). Demineralized water was used throughout the experiment.

### 2.2. Synthesis of L-Tyrosine-Conjugated-Chitosan (Tyr-c-CS)

The Tyr-c-CS preparation method is outlined in Figure 1. An amount of 500 mg CS flakes dissolved in 1 vol% of 50 mL acetic acid solution and stirred at room temperature until the homogenous solution was obtained in two separate flasks. To activate the carboxyl group of amino acid, 485 mg Tyr, NHS (1.2 mol equivalent to Tyr), and EDC (1.2 mol equivalent to Tyr) were dissolved in a dilute HCl acid solution via sonication. To one homogeneous solution of CS, activated Tyr solution and HOBt (1.2 mol equivalent to Tyr) were added and kept for stirring at 4 °C for 30 min followed by 24 h stirring at room temperature and another CS solution kept aside. The dialysis of the prepared Tyr-c-CS solution was performed against demineralized water in a dialysis tube (MW limit value 13 kDa) for three days. Both the solutions were centrifuged for 10 min at 10,000 rpm to remove all unreacted species.

### 2.3. Neat CS and Tyr-c-CS Membrane Fabrication

The solution casting method is applied to cast neat CS and Tyr-c-CS solutions using GARDCO micrometric film applicator (Paul N. Gardner, Columbia, MD, USA) onto the porous PES support. The membrane was placed in laminar airflow for 24 h, evaporating the solvent and forming a film. It was then dried in a hot air oven, ramping from ambient to 100 °C by 1 °C/min of heating rate, and it was held overnight. After cooling down to the ambient temperature, the membrane with a circular diameter of 45 mm was cleft at the flat sheet membrane module to accommodate the permeation of the test module, as shown in Figure 2.

### 2.4. Structural Characterization of Material and Membrane

Numerous characterization techniques were performed to characterize the resultant CS and Tyr-c-CS membranes, as discussed below:

To confirm the conjugation, the Fourier transform infrared spectroscopy (FTIR) (SHIMADZU, IR Affinity 1, Nakagyo-ku, KYO, Japan) technique was used, and the presence of chemical bonds in the Tyr-c-CS film was identified. The FTIR spectra were collected at room temperature at the frequency range of 400–4000 cm^−1^ using the attenuated total reflectance (ATR) mode. The crystalline or amorphous nature of the active layer of the prepared membrane was determined with Rigaku SmartLab (Akishima-shi, TYO, Japan) XRD using 9 kW power X-ray diffractometer with Cupper K_α_ radiation of λ = 1.54 Å in a 2θ range of 5–45° with a scan rate of 5° s^−1^. Top surface views and cross-sectional views of the prepared membranes were taken using field emission scanning electron microscopy (FESEM) (Zeiss Sigma, Dublin, CA, USA). The top surface of the PES support, CS, and Tyr-c-CS membranes was also captured in high resolution and analyzed by atomic force microscopy (AFM) (Innova Bruker, Camarillo, CA, USA). A scan rate of 1.0 Hz was preferred for tapping mode, and captured images were further processed using WSxM software to obtain an idea of the two-dimensional and three-dimensional structure of top surfaces, roughness, and height profiling. The weight loss analysis was performed on the active layer, using TG209 F1 Libra (Netzsch, Selb, BY, Germany) A thermogravimetric analyzer (TGA) was used to characterize the temperature stability. The analysis was performed with temperature ramping from 25 °C to 700 °C by 10 °C/min under a nitrogen atmosphere. Swelling ability tests of the active layer of the membranes were performed at different relative humidity (RH). The RH of the environment was maintained using 0 to 100 wt% solutions of glycerol. The mixture was poured into three mouth-round bottom flasks, as shown in Figure 3. The sample was hung using a binder clip and left in the humid atmosphere for 10–12 h through the central opening. Through one opening thermometer was inserted to measure the temperature, and through another, a humidity meter was inserted to measure the humidity.

The following relation was used to measure the swelling (%) of the membrane [51]:(5)$$S\%=\frac{M_{w}-M_{d}}{M_{d}}\times 100$$
where $M_{w}$ and $M_{d}$ are the masses of the humidified and dry membranes, respectively.

$M_{d}$ was measured after removing all moisture content from the humidified membrane.

### 2.5. Gas Permeation Setup

Figure 4 shows a diagrammatic representation of the designed gas permeation setup. The round-shaped thin-film membrane with an effective area of ~8 cm^2^ has been mounted in a counter-current membrane module. The influence of temperature was studied by placing the module in a cabinet equipped with a controllable thermostat with the possibility of regulating the temperature from ambient to 115 °C. The CO_2_/N_2_ as feed gas in a proportion of 20:80 and argon as sweep gas were sent at 35 and 32 L/h, respectively, via digital mass flow controllers (AALBORG, Orangeburg, NY, USA). Two high-pressure liquid chromatography (HPLC) pumps (Shimadzu, LC 20AD, Nakagyo-ku, KYO, Japan) were used to monitor the humidity, and two back pressure regulators (Swagelok, Fremont, CA, USA) were used to control the pressure of the membrane module. The flow rates of dehumidified exiting streams were monitored using high-precision gas flow meters (Agilent, Santa Clara, CA, USA) and collected at the water knockout points. Later, gas chromatography (GC Varian 450) was used to study the existing stream composition with a thermal conductivity detector (TCD). The analysis was performed by varying temperature, pressure, feed, and/or sweep side moisture flow rates. The readings were recorded at specified conditions and maintained until a steady state was attained. The effectiveness of a membrane was tested based on its CO_2_ permeance and CO_2_/N_2_ selectivity.

## 3. Results and Discussions

### 3.1. Spectroscopic Analysis

To draw information on the molecular structure change, which may happen due to the conjugation of CS, the Fourier transmission infrared (FT-IR) spectra (Figure 1a) and X-ray diffraction (XRD) spectra (Figure 1b) were recorded and evaluated for all the samples: CS powder, Tyr powder, CS film, and the Tyr-c-CS film. In Figure 1a, from 3100–3500 cm^−1^, the CS film exhibited a medium broad peak, and the Tyr-c-CS film exhibited a broad, but strong, peak. These peaks are assigned to hydroxyl and amine stretch [50]. The increment in the intensity of O-H and N-H peaks for Tyr-c-CS partially proves the successful grafting of Tyr onto CS (Figure 1) [52].

Additionally, in the FTIR spectrum in the C-H stretch region, the lower intensity peak shift is observed from 2887 cm^−1^ for CS film to 2951 cm^−1^ for Tyr-c-CS film, suggesting a possible interaction between CS and Tyr molecules. In the Tyr-c-CS film, the presence of Tyr was confirmed by a weak aromatic C-H peak at 1697 cm^−1^, a medium aromatic C=C peak at 1555 cm^−1^, and N-H bends at 1613 cm^−1^. An amide bond (C=O-N-H) peak formed due to the interaction of CS and Tyr molecules was observed at 1646 cm^−1^ in the Tyr-c-CS film spectrum. The strong C-O stretch at 1053 cm^−1^ was observed for the Tyr-c-CS film, as was seen for both powder CS and CS film [53]. Strong N-H bending vibrations of secondary amide are assigned for the peaks at 1559 cm^−1^ and 1599 cm^−1^. Due to CH_2_ scissoring, which occurred at 1465 cm^−1^ for CS, the band was broadened in conjugated CS. However, two new peaks were observed in the fingerprint region of the Tyr-c-CS film spectrum at 1394 cm^−1^ and 721 cm^−1^ due to phenolic OH and benzene ring, which further confirms the conjugation of chitosan with L-tyrosine [54,55].

Material crystallinity analysis is usually performed using the XRD technique. As shown in Figure 1b, the XRD pattern of CS powder demonstrates a weak and a strong diffraction peak that appeared at 2θ values of 10.25° and 21.21°, respectively. The initial peak indicates the amorphous region, and the second peak represents the crystalline structure. Therefore, as expressed in the reported literature, CS creates a semi-crystalline structured polymer [56]. Unlike the CS powder, the neat CS film exhibited a lower amplitude peak in the 2θ range of 10–20°. This suggests an increase in disarray in chain alignment compared to the CS powder. As perceived in Figure 1b, Tyr-c-CS film exhibited comparatively broader hump with low-intensity peaks, which is an indication of a highly amorphous region due to the conjugation of Tyr [57].

### 3.2. Thermal Stability Analysis

A thermal stability test of CS powder, Tyr powder, pristine CS active layer, and Tyr-c-CS active layer was performed using the thermogravimetric analyzer (TGA). Specimens 5–10 mg were heated at 10 °C/min in nitrogen flow post-evacuation in ceramic cans. The weight loss profiles for each of them are shown in Figure 2. As observed from the TGA spectra shown in Figure 2a, the temperature at the onset of thermal degradation is ~300 °C for Tyr powder and 244 °C for CS powder [58,59]. In Figure 2b, thermal degradation trend is shown for both the prepared membranes. The CS membrane showed 8.21% loss between 25–114 °C in the first stage due to the removal of unbound free moisture, followed by 15% loss up to 227 °C in the second stage due to the evaporation of bound free moisture content. In the third stage, the membrane experienced a sudden weight drop from 15% to 55% between 215–358 °C, which can be ascribed to the thermal degradation of the polymer structure. On the other hand, the Tyr-c-CS membrane represents only 5% loss up to 212 °C and later experienced a sudden weight drop of 30% up to 250 °C followed by 48% weight loss up to 410 °C in the third stage. It has been observed that the thermal stability has a trend following the order of Tyr > CS > Tyr-c-CS membrane > CS membrane. Apart from this, the weight loss is lower in the Tyr-c-CS active layer by 8% compared to the CS active layer. Hence, CS modified with Tyr via chemical conjugation has shown better thermal stability, which can be ascribed as the advantage of using amine conjugation instead of amine blending compared with available literature [60]. The amide bond formation upon conjugation of the two moieties resulted in the increased chemical stability of the molecule, which in turn enhanced thermal stability. According to the TGA analysis, the fabricated Tyr-c-CS membrane is well suited for CO_2_ separation processes and CO_2_ capture from post-combustion flue gas due to its comparable high thermal stability.

### 3.3. Morphological Analysis

The cross-sectional and surface images of the PES support and fabricated Neat CS and Tyr-c-CS membranes are shown in Figure 3. The commercial PES support provides mechanical strength to the CS and Tyr-c-CS active layer in the harsh environment, which is required for the CO_2_/N_2_ gas separation. Moreover, the porous structure (average pore size ~30 nm) imparts resistance-free transport of gas molecules across the membrane after penetrating through a dense active layer (Figure 3a). The formation of defect-free dense active layers of CS and Tyr-c-CS onto PES support required for facilitated transport of CO_2_ molecules along with solution–diffusion [61] is confirmed in Figure 3b,c. The cross-sectional view of the fabricated membrane (Figure 3d,e) clearly shows the active layer formed with a thickness of 500–700 nm on PES support. The cross-sectional view indicates no pore filling of PES support by active layer solution [16]. The pore filling of the support layer is undesirable, as it increases the effective layer thickness and reduces the permeance of the gas molecules.

A more detailed surface analysis of the CS membrane and the Tyr-c-CS membrane was conducted using atomic force microscopy (AFM). The typical two-dimensional, three-dimensional, and height profiling of the membrane image obtained from the AFM analysis is shown in Figure 4. The images represent distinct peak and valley regions. The average roughness of the CS membrane was 5.72 nm, whereas Tyr-c-CS was 10.22 nm. The enhanced roughness in Tyr-c-CS is due to the conjugation of amino acid in the polymeric chain of CS, which is crucial for CO_2_ separation, as it may increase the effective area for molecular transport [62].

### 3.4. Moisture Retention Test

The cardinal importance of the swelling parameter in facilitated transport can be understood with the zwitterion mechanism. Moisture in the membrane can accelerate the CO_2_ uptake capacity of carrier amine by two-fold (Equations (1)–(4)). It induces flexibility in the polymer matrix via the plasticization effect and contributes to the enhancement of free volume by changing the intermolecular structure of the membrane [63]. Hence, in addition to the facilitated transport mechanism, CO_2_ molecules can also pass by following the solution–diffusion mechanism through the swollen membrane. As depicted in Figure 5, the moisture holding capacity of the fabricated membranes follows two types of trends: up to 75% relative humidity (RH), they showed linear increment with lesser slope, and with respect to 80–100% RH, they showed linear increment with a steeper slope. The RH at which the slope positions change can be called critical relative humidity. Beyond this point, the humidity present in the feed gas induces chain relaxation and determines the swelling of the membrane [64,65].

### 3.5. Gas Permeation Study

#### 3.5.1. Effect of Temperature on the Separation Efficiency of Dry Membrane

Initially, the study of the CO_2_/N_2_ separation performance of the Tyr-c-CS membrane was performed in dry conditions in the temperature range of 25 °C to 115 °C, as shown in Figure 6a,b. As expected, the CO_2_/N_2_ selectivity was increased permeance by ~four-fold, from 1.2 to 5.2, and CO_2_ permeance by ~three-fold, from 7.9 GPU to 27 GPU for Tyr-c-CS membrane, when the temperature raised from room temperature to 85 °C. Whereas, the neat CS membrane showed a CO_2_ permeance of 16 GPU and CO_2_/N_2_ selectivity of 4.3 under similar operating conditions, as shown in Appendix A. The gas permeability (P) is a product of the contribution of both the diffusion and sorption of the gas molecules in the membrane.
P = D × S(6)
where D and S are the diffusivity and solubility coefficients, respectively [8].

At lower temperature, the sorption effect dominates due to higher affinity of CO_2_ towards -NH_2_, forming the carbamate, while the diffusion of the gas molecules is much less, resulting in low CO_2_ permeance of the membrane at lower temperature. With increase in temperature (up to 85 °C), the diffusion of CO_2_ through the membrane advances owing to the increase in diffusion coefficient (Fick’s law) and flexibility of the membrane with temperature. This causes the enhancement in the CO_2_ permeance of the membrane up to 85 °C. Beyond 85 °C, the sorption effect reduces due to the dominance of backward reaction of carbamate formation, resulting in less solubility of the gas into the membrane. Due to this effect, the reduction in the solubility at higher temperature (>85 °C), there occurs a sharp decrease in the CO_2_ permeance of the membrane at higher temperature.

Facilitated transport membranes use a carrier molecule to selectively transport CO_2_ over N_2_ by binding to CO_2_ and facilitating its transport across the membrane. The effect of temperature on CO_2_/N_2_ selectivity may differ between facilitated transport membranes and traditional membranes that rely on gas diffusion rates [66]. In CS and Tyr-c-CS facilitated transport membranes, CO_2_/N_2_ selectivity increases with temperature up to a certain point. This is because the carrier molecule used in the facilitated transport mechanism can have temperature-dependent binding properties for CO_2_, which can impact the overall CO_2_ transport rate [30]. The reaction of CO_2_ with amine (-NH_2_) functional group is a reversible exothermic reaction. Thus, up to a certain temperature, the forward reaction dominates forming the carbamate, while, at higher temperature, the backward reaction predominates, resulting in the decrease in the carbamate formation [67]. The permeance vs. temperature plot, when increasing (25–115 °C) and decreasing (115–85 °C), also incorporated into the supporting information Appendix A, demonstrates the stability and the reusability of the dry Tyr-c-CS membrane for gas separation applications.

#### 3.5.2. Effect of Temperature on the Separation Efficiency of Swollen Membrane

In practice, moisture is one of the constituents of flue gas. Thus, higher efficiency of the membrane is much more needed under swollen conditions. The effect of temperature on CO_2_ and N_2_ gas mixture on the separation performance of the conjugated polymeric membrane under swollen condition was investigated from 25–115 °C oven temperature and 32 psi feed absolute pressure under controlled moisture flow of 0.03 mL/min, as well as 0.05 mL/min in the feed and sweep sides, respectively. As depicted in Figure 6c,d, the CO_2_ permeance increased from 32 GPU to 103 GPU, and CO_2_/N_2_ selectivity increased from 9 to 31.3 when the temperature increased from 25 to 85 °C. Whereas, the neat CS membrane showed the CO_2_ permeance of 60 GPU and CO_2_/N_2_ selectivity of 21 at 85 °C temperature, as shown in Appendix A. The CO_2_ is separated from the CO_2_/N_2_ mixture via the facilitated transport mechanism. The amine functional group-rich membrane (Lys-c-CS) will interact with more molecules of incoming CO_2_ in the presence of water because the zwitterion mechanism, as described by Caplow [42], would follow reactions (3) and (4). Thus, it can be concluded that, in the presence of moisture, the rate of the CO_2_ adsorption on the membrane will be enhanced owing to the 1:1 CO_2_-NH_2_ interaction, which is theoretically the maximum CO_2_ loading of amines in the presence of a water molecule. As the zwitterion mechanism is a reversible formation reaction, the adsorbed CO_2_ will become desorbed, and it undergoes immediate reaction with the adjacent -NH_2_ group of the Lys-c-CS polymer and eventually travels across the membrane via a facilitated transport mechanism (simultaneous reversible carbamate formation) and, thus, is separated out from the gas mixture. This enhancement in CO_2_ permeance and selectivity can be attributed to the fact that the conjugation of chitosan provided additional free amines that actively facilitated transport reaction and promoted carbamate formation in the presence of water molecules [68,69]. In the case of N_2_ transport, similar to the trend in dry conditions, there was not much change observed with an increase in temperature. With further increases in temperature from 85–115 °C, the separation performances declined due to the reduction of sorption of gas molecules and its diffusion through the membrane due to the deterioration of the moisture-holding capacity at higher temperature ranges [70]. The permeance vs. temperature plot, which was increasing (25–115 °C) and decreasing (115–85 °C), was also incorporated into the supporting information, as depicted in Appendix A, which demonstrates the stability and the reusability of the swollen Tyr-c-CS membrane for the gas separation applications.

#### 3.5.3. Effect of Sweep Side Humidity on the Separation Efficiency of Swollen Membrane

The effect of moisture flow on the sweep side on the CO_2_ separation performance of the membrane was studied in the range of 0–0.09 mL/min at an absolute pressure of 32 psi for feed, the temperature of 85 °C, and feed side moisture of 0.03 mL/min. The conditions were chosen based on the study conducted on the chitosan/silk fibroin membrane by our previous group [17]. As shown in Figure 7a,b, for 0 mL/min moisture flow rate on the sweep side, CO_2_ permeance and selectivity were 27.9 GPU and 5.2, respectively. On the other hand, at 0.05 mL/min sweep moisture flow rate, the CO_2_ permeance and selectivity were amplified by ~267% and ~496%, respectively, and reached to 103 GPU of CO_2_ permeance and selectivity of 31.3. The N_2_ permeance was reduced from 5.5 GPU to 4 GPU with an increase in moisture content in the sweep side from 0 to 0.05 mL/min. As observed in Figure 7a,b, the separation performance of the membrane improved with increasing moisture flow on the sweep side and then decreased after 0.05 mL/min. The findings suggest that water plays a beneficial role in CO_2_ separation by aiding in the formation of CO_2_-carrier complexes. The reaction between CO_2_ and the carrier leads to the formation of bicarbonate (HCO_3_^−^) in the high-pressure feed region, which dissociates in the low-pressure permeate region to release CO_2_ and water. The diffusivity of the HCO_3_ complex is greater than that of CO_2_ alone, which improves the CO_2_ flux or permeance compared to non-reacting gases, such as N_2_. Water-induced swelling of the membranes increases the free volume and reduces mass transfer resistance to gas molecules. Water also acts as a plasticizing agent, inducing chain relaxation and increasing membrane flexibility. CO_2_/N_2_ selectivity increases with sweep water flow rate up to 0.05 mL/min, beyond which further improvement is not observed due to carrier saturation [68,71]. At 0.07 mL/min and more sweep-side moisture flow, due to excessive swelling effect mass transfer resistance to N_2_, gas decreases further. As a result, the CO_2_/N_2_ selectivity decreased at higher sweep moisture flow and, due to carrier saturation phenomena, CO_2_ permeance decreased.

#### 3.5.4. Effect of Feed Pressure on the Separation Efficiency of Swollen Membrane

To investigate the effect of pressure difference across the membrane, gas permeation tests were conducted at the different feed pressures of 32 psi, 44 psi, 59 psi, and 74 psi at an operating temperature of 85 °C, sweep moisture flow rate at 0.05 mL/min, and feed moisture flow rate at 0.03 mL/min. For binary CO_2_/N_2_ gases, as shown in Figure 7c,d, CO_2_ permeance and selectivity decreased from 105 GPU to 66 GPU and 31 to 7.1, respectively, when feed pressure increased from 32 psi to 74 psi in the membrane. As known, at high pressure, the facilitated transport of CO_2_ leads to the formation of complexes with amino groups. This increased formation of complexes inhibits the interaction of active sites with coming CO_2_ molecules, leading to reduced CO_2_ separation performances [30]. Thus, the decreased CO_2_ permeance and selectivity are due to the carrier saturation phenomena that hamper the efficiency of facilitated transport of CO_2_ molecules.

#### 3.5.5. Prolonged CO_2_/N_2_ Gas Separation Test of Tyr-c-CS Membrane

Durability under harsh prolonged conditions for separation performance is a vital necessity to be an efficient membrane. CO_2_/N_2_ gas separation performance of the fabricated Tyr-c-CS membrane was analyzed for over 150 h at 85 °C temperature, 32 psi feed pressure, and feed to sweep side moisture flow ratio of 1.67. Due to the covalent link formed between the molecules of chitosan and tyrosine during conjugation, the membrane retains its CO_2_ separation performance over a period of 150 h with barely any fluctuation, exhibiting the remarkable stability of the membrane performance, as shown in Figure 8. After the gas separation test, the Tyr-c-CS membrane was demounted from the module, dried in an oven, and tested with various analytical techniques, such as FESEM, XRD, FTIR. AFM, etc. and shown in Appendix A. The FT-IR and XRD spectra of the Tyr-c-CS membrane (after stability test) confirmed that the crystal and electronic structure remained intact. The FESEM and AFM images inferred the constancy of the surface features. The obtained results demonstrate the stability of the membrane after passing through harsh environment provided during the experiment. Thus, we can say that the membrane is stable and re-usable.

To better understand the upgraded performance of the fabricated Tyr-c-CS membrane and to support the utilized strategy of conjugation of amino acid onto chitosan polymer, we have compared the obtained results from this study with various polymeric membranes (Table 1).

#### 3.5.6. Robeson Upper Bound Curve

Robeson’s upper bound curve illustrates the trade-off between CO_2_ permeability and CO_2_/N_2_ selectivity on a logarithm plot for polymeric membranes. CO_2_ permeability is reported in Barrer units and can be calculated by multiplying permeance with active layer thickness. The results are meticulously compared with the well established contemporary studies on gas separation with FTM, presented in Table 1. In this comparative study, the performance of the fabricated membrane is evaluated from the Robeson upper bound curve (2008) and modified Robeson upper bound curve (2019) [75] (Figure 9) and the other available literature [16,17,44,72,73,74]. The curve clearly indicated that, when compared to neat CS membrane, the Tyr-c-CS membrane showed better separation performance in both dry and wet conditions. This is due to the fact that conjugation of tyrosine onto the CS matrix enhanced the available amine active sites and also increased the roughness of the membrane. A pure CS membrane has a rather smooth surface, but when Tyr is conjugated, the topology of the membrane surface is significantly changed [76,77]. Little projecting bumps with a height of 10–30 nm are created, as seen in Figure 4, enhancing the membrane’s surface roughness and contributed in high CO_2_ permeance [78]. Moreover, Tyr-c-CS is a more stable membrane than other amine blended CS membranes due to the conjugation, so, in the future, if additional amine is blended to Tyr-c-CS, then it might show a significant result.

## 4. Conclusions

A high-performing Tyr-c-CS membrane was fabricated and analyzed in the presented work for a flue gas separation study. The physical and chemical properties of the fabricated membrane were well characterized utilizing various techniques. This work highlights the remarkable effect of amino acid conjugation on polymer backbones and its effect on enhanced CO_2_ separation. The amine groups in the conjugated L-tyrosine auspiciously attract CO_2_ molecules, but not N_2_, owing to the zwitterion mechanism for carbamate formation. This research also postulated a unique methodology to amplify the CO_2_ separation performance of the membrane by inducing plasticization and swelling, a commonly observed phenomenon during pressure and moisture application in such studies. The synthesized conjugated membrane, with an average thickness of 600 nm, showed high CO_2_ permeance of 103 GPU and reasonably good CO_2_/N_2_ selectivity of 31 at 85 °C and 32 psi absolute feed pressure. The facilitated transport and the solution–diffusion mechanism, along with the cumulative impact of carrier amines from L-tyrosine, enhanced the performance of Tyr-c-CS matrix.

## Data Availability

Data is unavailable due to privacy or ethical restrictions.

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
