# Peer review of "A Strategical Improvement in the Performance of CO2/N2 Gas Permeation via Conjugation of L-Tyrosine onto Chitosan Membrane"

_membranes, 2023, doi:10.3390/membranes13050487_

Round 1
Reviewer 1 Report
In this manuscript, a chitosan membrane modified with L-tyrosine was reported for CO2/N2 separation. The authors characterized the modified membrane with FTIR, XRD, TGA, SEM, and AFM, and evaluated the CO2/N2 separation performance of the modified membrane under different conditions. Overall, this manuscript is within the scope of the journal and could be considered for publication. I only have a minor comment. As shown in Figure 8, although the modified membrane showed better performance compared with the pristine chitosan membrane, the CO2/N2 separation performance of L-tyrosine modified membrane needs to be further improved to surpass the Robeson upper bound. For practical application, how could the performance of L-tyrosine modified membrane be further improved (further decreasing the thickness of the coating layer on PES membrane or other ways to improve the gas permeance and selectivity)? The author may add some discussion in the manuscript.
Author Response
Please see the attached response.

Reviewer 2 Report
I advise the authors to revise the manuscript based on the following comments.
Most of the figures provided in this manuscript are of poor resolution. This is not acceptable for publication.
Besides, authors should discuss their results (FTIR, SEM, etc) by comparing with the literature data.
Some key characterizations are missing. For instance, NMR should be performed to confirm the successful formation of the structure (Tyr-c-CS, Scheme 1).
A prolonged separation test should be carried out to show the stability of the membranes.
Other comments are as follows.
Abstract – The selectivity of CO2/N2 should be highlighted.
Introduction – Authors should review the commercial membranes made of chitosan for gas separation.
Figure 3 – I advise the authors to supply the the entire cross-sectional image of the membrane.
Figure 6 (a) and (c) – Please explain why the operating temperature testing is not consistent for both membranes.
Table 1 – Authors should include if the feed gas is single gas or mixed gas. If it is mixed gas with different composition, the permeance and selectivity will be affected. All these must be discussed properly.
Section 3.6.5 – What are the key message the authors wanted to deliver based on this figure? The statement is incomplete!
Author Response
Please see the attached response.

Reviewer 3 Report
Please see the attached file.

Author Response
Please see the attached response.

Round 2
Reviewer 2 Report
Authors had done a good job in revising the manuscript based on my comments. I therefore recommend for its publication.
Author Response
I would like to thank the reviewer for acknowledging our work and considering the revised manuscript for submission. We have modified the English and checked for spelling mistakes.
Reviewer 3 Report
Please see attached file.

Author Response
"Please see the attachment."
